# A Survey of COVID-19 Diagnosis Using Routine Blood Tests with the Aid of Artificial Intelligence Techniques

**DOI:** 10.3390/diagnostics13101749

**Published:** 2023-05-16

**Authors:** Soheila Abbasi Habashi, Murat Koyuncu, Roohallah Alizadehsani

**Affiliations:** 1Department of Computer Engineering, Atilim University, 06830 Ankara, Turkey; 2Department of Information Systems Engineering, Atilim University, 06830 Ankara, Turkey; murat.koyuncu@atilim.edu.tr; 3Institute for Intelligent Systems Research and Innovation (IISRI), Deakin University, Waurn Ponds, Geelong, VIC 3216, Australia

**Keywords:** COVID-19, blood tests, RT-PCR, machine learning, deep learning

## Abstract

Severe Acute Respiratory Syndrome Coronavirus 2 (SARS-CoV-2), causing a disease called COVID-19, is a class of acute respiratory syndrome that has considerably affected the global economy and healthcare system. This virus is diagnosed using a traditional technique known as the Reverse Transcription Polymerase Chain Reaction (RT-PCR) test. However, RT-PCR customarily outputs a lot of false-negative and incorrect results. Current works indicate that COVID-19 can also be diagnosed using imaging resolutions, including CT scans, X-rays, and blood tests. Nevertheless, X-rays and CT scans cannot always be used for patient screening because of high costs, radiation doses, and an insufficient number of devices. Therefore, there is a requirement for a less expensive and faster diagnostic model to recognize the positive and negative cases of COVID-19. Blood tests are easily performed and cost less than RT-PCR and imaging tests. Since biochemical parameters in routine blood tests vary during the COVID-19 infection, they may supply physicians with exact information about the diagnosis of COVID-19. This study reviewed some newly emerging artificial intelligence (AI)-based methods to diagnose COVID-19 using routine blood tests. We gathered information about research resources and inspected 92 articles that were carefully chosen from a variety of publishers, such as IEEE, Springer, Elsevier, and MDPI. Then, these 92 studies are classified into two tables which contain articles that use machine Learning and deep Learning models to diagnose COVID-19 while using routine blood test datasets. In these studies, for diagnosing COVID-19, Random Forest and logistic regression are the most widely used machine learning methods and the most widely used performance metrics are accuracy, sensitivity, specificity, and AUC. Finally, we conclude by discussing and analyzing these studies which use machine learning and deep learning models and routine blood test datasets for COVID-19 detection. This survey can be the starting point for a novice-/beginner-level researcher to perform on COVID-19 classification.

## 1. Introduction

SARS-CoV-2 was first recognized in China, after which the severe pneumonia yielded by the virus, called COVID-19, rapidly circulated worldwide [1,2]. COVID-19 has different clinical symptoms, such as dyspnea, fever, cough, myalgia, fatigue, gastrointestinal complications, and headache [3,4]. This virus is risky and affects the mortality of individuals with compromised immune systems. Medical professionals and infectious disease experts from the entire globe are seeking a solution for the disease. COVID-19 has been the primary source of death in many countries around the world, with the United States, Italy, and Spain having one of the highest number of deaths [5]. Figure 1 demonstrates the global 14-day COVID-19 case notification rate per 100,000 as of 15 July 2020.

To date, a couple of diagnostic techniques have been adopted by physicians, such as RT-PCR [6,7], imaging resolutions and blood checks [8]. RT-PCR, which is the best measure for the analysis of COVID-19 [9,10], tolerates a low sensitiveness (60–71%), more extended waiting duration for the results [11], and poses extra responsibilities to the healthcare system, demanding pricey devices [12,13,14]. Additionally, there is a shortage of testing kits, reagents, and trained personnel for analysis, particularly in less-developed nations [12]. Thus, scientists have searched for extra reachable techniques of diagnosis, among which imaging techniques have obtained remarkable attraction [15]. Chest CT and chest X-rays are mainstream imaging alternatives for diagnosing COVID-19 infection. Chest CT presents a better sensitivity [16], similar to RT-PCR [12]; however, it also has numerous downsides, such as hospital-acquired infection, radiation safety, and lower access rates to CT devices [17,18,19]. Chest X-ray is a less costly imaging choice and uses a lower radiation dose than CT, while virtually every health center and most clinics will have access to X-ray equipment [20,21,22]. Chest-X-ray, similar to CT, affords medical physicians with imaging symptoms of SARS-CoV-2 contamination, e.g., ground-glass opacity, but suffers from an increased rate of false-negative results [23,24,25]. Blood tests are broadly accessible and have much lower costs than RT-PCR and imaging tests. Since biochemical parameters contained in ordinary blood exams, for example, lactate dehydrogenase (LDH), C-reactive protein (CRP), etc. [26], change over the course of the COVID-19 contamination, blood tests can provide physicians with data about the diagnosis of COVID-19 [12]. Consequently, blood tests may additionally supply a potentially precious instrument for the quick screening of infected patients and compensate for the deficiency of RT-PCR and CT scans by providing an initial step of detection [27].

The health industry is impatiently following new techniques and technologies to address the growth of the COVID-19 epidemic in the international health crisis. AI is known to be one of the grandest uses of global technology that can follow the speed and detect the growth rate of COVID-19 and determine the risk and severity of COVID-19 patients. AI can also predict cases of death by adequately analyzing previous patient data. Moreover, AI can help us battle this virus by testing people and providing medical assistance, data and information, and recommendations regarding disease control [28].

Machine learning (ML) [29,30,31], as a class of AI, includes the algorithmic modeling structures of statistical models and only requires a small amount of knowledge to learn how to handle problems [32,33,34]. On the other hand, deep learning (DL) [35] is a class of ML that concentrates on making deep structural neural-network-based models that learn from data utilizing feed-forward and back-propagation. DL has improved significantly in the last two decades in several activities [36,37,38]. However, it needs a vast amount of data to learn. Exceptional cases of DL, where large-scale datasets are not required to train, have been generative models and transfer learning [39,40,41].

This survey introduces a series of main works of AI containing ML and DL research articles on COVID-19 diagnosis using routine blood tests. In total, we investigated 92, of which 82 articles are ML-based, and the rest are based on DL models. Because of the rapid growth of COVID-19 cases, we have quoted many published articles before conducting a thorough investigation, so these articles should be analyzed for their accuracy and quality in peer review.These articles are classified in two tables, in which, by comparing different works, it is determined which AI algorithms or performance metrics are used more to detect COVID-19, as well as which blood test datasets are used.

There are four main sections in this study. Section 2 discusses the procedure followed for selecting the research articles. A summary of contemporary machine learning and deep learning research is given in Section 3. Section 4 presents the blood features used in previous articles. The approaches are examined, and the outcomes of various models are discussed in Section 5. In Section 6, the article is concluded.

## 2. Protocol for Choosing COVID-19-Related Research Articles

To choose the research articles, the most pertinent keywords, such as COVID-19, routine blood tests, machine learning, and deep learning, were used. Moreover, we employed digital databases, including IEEE Xplore, Elsevier, Springer, and MDPI, to collect only English-language literature. Note that research articles about blood tests and rapid antigen tests were not reviewed. Figure 2 shows the details of the statistics of ML and DL publications on COVID-19. The search strategy was adjusted to obtain the maximum number of documents. Table 1 lists the number of documents retrieved by the queries referenced. The results were thoroughly scrutinized to seek relevant results. The queries yielded 4785 documents.

The IEEE database was subject to a review of 620 studies that were evaluated according to their titles and abstracts. This screening led to the exclusion of 459 documents, as their material was not related to the analysis, and the remaining 161 were chosen for a full-text review. After a full-text review of the studies, 148 were rejected, and the remaining 13 were included in the present review. In total, the titles and abstracts of 1123 studies from the Springer database were scanned through. After the screening process, 992 documents were deemed unsuitable and only 131 were qualified for a full-text review. In total, 112 of the studies were disregarded and 19 were ultimately included in this study. The titles and abstracts of 1201 studies in the Elsevier database were examined. After the screening was finished, 1102 documents were disregarded because they had no association with the investigation, leaving 99 documents to be analyzed for a full-text review. From these studies, 21 were accepted and 78 rejected. A review of the MDPI database was conducted, resulting in 337 studies, with the evaluation based on their titles and abstracts. The screening process resulted in the elimination of 301 documents, as they were not relevant to the analysis, while 36 were selected for a full-text review. A complete textual analysis of the studies led to the rejection of 28, while the remaining 8 were included in the current review. Finally, the titles and abstracts of 1504 studies in other databases were evaluated. Following the screening, 178 documents were kept for a full-text review, since the other 1326 had no association to the topic being studied. Out of these studies, 147 were rejected and 31 were used for the final analysis.

This section delineates the content of the 92 studies found in the databases investigated. The results of the research article search are summarized in Figure 3.

## 3. Overview of Machine Learning and Deep Learning Methods

Routine blood tests can be used to quickly diagnose COVID-19 infection utilizing AI-based techniques such as ML and DL. These models can uncover potential connections between different qualities in blood test results and provide information to guide decisions.

This section contains research articles that use routine blood tests to diagnose COVID-19 while taking into account ML and DL models. The majority of COVID-19 detection techniques used are shown in Figure 4.

### 3.1. Machine Learning

More focus has been placed on the potential use of ML methods to address COVID-19 diagnosis using routine blood testing. The following are some well-known ML algorithms that have been applied for COVID-19 diagnosis [42].

Support vector machine (SVM) [43] is used for classification. The goal of SVM is to identify the best hyperplane for differentiating the features. There are numerous ways to draw the hyperplane and an ideal one has been discovered that best separates the dataset. SVM is a highly accurate model, and it is very unlikely that overfitting will occur.

Random Forest (RF) [44] is another ensemble technique using decision tree that utilizes a re-sampling process called the bagging method that creates multiple trees and handles weak classifiers in a different way. RF is effective for highly complex problems and can handle missing data and unbalanced datasets.

K-nearest-neighbor (KNN) [45] is a classification algorithm with which a sample’s label can be predicted using the labels of its closest neighbors. It is necessary to choose the parameter K and use the attribute-distance computation metric to determine which other data points are the closest neighbors.

Logistic regression (LR) [46] is a classification algorithm. Based on the value of independent features, LR models the probability of samples belonging to a particular class. Then, the model can be used for predicting the probability that a given sample belongs to a certain class. LR has simple calculations and can support continuous numerical values, while non-linear data cannot be handled by LR.

Decision tree (DT) [47] is a Supervised ML algorithm. To characterize the connections between attributes and a class label, DT generates a tree-structured model. It divides observations recursively based on the property with the highest gain ratio value that is the most informative. The data are divided in the nodes and decisions are made in the leaves.

Naive Bayes (NB) [48] is a simple probabilistic classifier based on Bayes’ theorem. It cannot handle missing data.

Extreme Gradient Boosting (XGBoost) [49] is a machine learning algorithm based on decision trees, which employ a gradient boosting structure. XGBoost is a library of machine learning which uses a scalable and distributed form of Gradient-Boosted Decision Tree (GBDT). There are many advantages to using this machine learning library, such as parallel tree-boosting, and it is the leading machine learning library for regression, classification, and ranking.

LGBM (LightGBM) [50] is an efficient, distributed, and high-performing gradient boosting framework based on decision tree algorithms, used for ranking, classifying, and other machine learning processes. LGBM is an application of the gradient boosting framework that is based on tree-structured algorithms.

Table 2 displays the most recent machine learning models for early detection of COVID-19 or assessing the disease severity level of COVID-19 patients based on laboratory and clinical data.

### 3.2. Deep Learning

The performance enhancements of hardware components, such as graphics cards, and the drop in unit costs are two reasons DL has grown in popularity. DL has also been aided by machine learning and information processing studies [51,52,53], as well as an increase in training data. Numerous domains, including computer vision [36,54,55,56], natural language processing [57,58,59,60,61,62], and speech recognition [63], have extensively used various deep learning architectures, such as ANN, CNN, and RNN. ANN is a method of processing information that draws inspiration from the organic nervous system of humans. This structure is made up of neurons, activation processes, and input, output, and hidden layers. In an ANN, each layer comprises a hierarchy of neurons. The input for the following layer is the layer’s output before it. From the incoming data, each layer learns increasingly intricate relationships. A deep learning system called CNN was created to analyze visual data, such as photographs and movies. Different layer types perform various functions on CNN. The names of these components are the convolution layer, pooling layer, fully connected layer, and activation function layer. In RNN structures, the outcome is influenced by both the other and current inputs. These networks produce their results by combining data from the past and present.

In this study, we selected 11 deep-learning-based studies, shown in Table 3. Overall, the number of works presented based on deep models is small; however, their accuracy in datasets with a large number of data is superior to ML methods.

**Table 2 diagnostics-13-01749-t002:** List of ML models that diagnose COVID-19.

Ref.	Dataset Source	Dataset Size (COVID-19)	Total Features (Selected)	Model Used	Metric Results
[64]	Lanzhou Pulmonary Hospital	253 (105)	49 (11)	RF	Acc: 95.95%, Sen: 95.12%, Spe: 96.97%
[65]	Hospital Sirio Libanes	1945 (545)	40 (40)	RF, DT, XGBoost, GBM, LGBM, SVM, ANN, KNN	Acc: 90%
[66]	Hospital Israelita Albert Einstein, Brazil	5644 (1128)	50 (17)	SVM, LR, RF, DT, KNN	Acc: 89.78%
[9]	San Raphael Hospital in Italy	1624 (844)	72 (34)	NB, LR, RF, KNN, SVM	Acc: 74%, Sen: 70%, Spe: 79%, AUC: 74%
[67]	Premier Healthcare Database	2183 (1020)	29 (15)	XGBoost	Sen: 95.9%, AUC: 91%
[16]	Taizhou hospital in China	114 (32)	14 (14)	LR, NB, IBk, DT	Acc: 84.21%
[68]	-	55,676 (1564)	12 (10)	GBM, MLP, RF, DT, KNN, LR, SVM, XGBoost	Sen: 93%, AUC: 91%
[69]	Rennes academic hospital	536 (106)	- (-)	RF, LR, ANN	AUC: 93%
[70]	San Raffaele Hospital in Italy	279 (177)	15 (15)	CNN and 15 supervised ML algorithms	Acc: 99.28%, AUC: 98.80%
[71]	-	300 (137)	8 (8)	SVM, LR, XGBoost	Acc: 87%
[72]	Hospital Israelita Albert Einstein, Brazil	5644 (1128)	13 (12)	RF, LR, XGBoost, SVM, KNN	Acc: 91%, Sen: 94%, Spe: 71%, AUC: 91%
[73]	University Medical Center, Ljubljana, Slovenia	5333 (160)	117 (35)	RF, ANN, XGBoost, SVM	Sen: 81.90%, Spe: 97.90%, AUC: 97%
[74]	Routinely collected laboratory, clinical, and demographic data	1040 (-)	5 (-)	ANN, extra trees, RF, XGBoost, catboost	Sen:92%, Spe: 82%, AUC: 92%
[75]	Hospital Israelita Albert Einstein, Brazil	608 (84)	108 (16)	RF, DT	Acc: 88%, Sen: 66%, Spe: 91%, AUC: 86%
[76]	Hospital Israelita Albert Einstein, Brazil	5644 (559)	108 (24)	NB, MLP, DT, BN, SVM	Acc: 95.15%, Sen: 96.80%, Spe: 93.60%
[77]	Three datasets of routine laboratory blood tests	2503 (1043)	Many (-)	SHAP, ELI5, LIME	Sen: 92%, Spe: 82%, AUC: 92%
[78]	Hospitals in Zhejiang, China	912 (361)	31 (10)	SVM, LR, RF, DT	Acc: 91%, Sen: 87%, Spe: 95%, AUC: 95%
[79]	-	398 (-)	42 (19)	XGBoost, SOM	Sen: 92.50%, Spe: 97.90%
[80]	West China Hospital, China	620 (211)	19 (19)	Multivariate LR	AUC: 87.2%
[81]	11 regions in China	659 (-)	Many (-)	DT	Acc: 89%, AUC: 88%
[82]	SMART hospitals	- (-)	- (-)	SVM, RF, NB	Acc: 93.33%
[83]	Hospital Israelita Albert Einstein, Brazil	5644 (2210)	Many (-)	LDA	Acc: 99.60%, Sen: 98.72% Spe: 98.99%, AUC: 99.38%
[84]	Hospital Israelita Albert Einstein, Brazil	598 (81)	108 (14)	RF, LR, ANN	Acc: 81%-87%, Sen: 43%-65%, Spe: 81%-91%
[85]	Hospital Israelita Albert Einstein, Brazil	5644 (558)	Many (-)	ANN, RF, Shallow learning	AUC: 95%
[86]	Hospital Israelita Albert Einstein, Brazil	5644 (65)	Many (-)	Er-CoV	Sen: 70%, Spe: 85%, AUC: 86%
[87]	Kepler University Hospital	1357 (653)	28 (-)	RF	Acc: 86%, AUC: 74%
[88]	Three Brazilian hospitals	1521 (-)	130 (-)	Adaboost, SVM, XGBoost, RF	Sen 96%, Spe: 93%
[89]	Three Brazilian hospitals	1521 (-)	130 (-)	HUST-19	Acc 94%
[90]	Oxford University hospitals	1157 (349)	- (-)	Different ML classifiers	Sen: 77%, Spe: 95% AUC: 93%
[91]	Five hospitals in New York	4098 (-)	Many (-)	XGBoost	AUC: 89%
[92]	Hospital Israelita Albert Einstein, Brazil	5644 (559)	108 (18)	RF, KNN, SVM, ET	Acc: 99.88%, Sen: 98.72%, Spe: 99.99%, AUC: 99.38%
[20]	San Raffaele Hospital in Italy	279 (177)	-(15)	KNN, ET, LR, DT, NB, RF, SVM	Acc: 82–86%, Sen: 92–95%
[93]	The RT-PCR COVID-19 test on the basis of routinely acquired blood tests	127,115 (1573)	Many (100)	SNN, KNN, LR, SVM, RF, XGBoost	AUC: 95%
[94]	New York Presbyterian Hospital	3346 (1394)	685 (33)	DT, LR, RF, GBDT, DT	Sen: 75.80%, Spe: 80.20%, AUC: 85.30%
[95]	Stanford Health Care, CA, USA	390 (31)	-(4)	LR	Sen: 86%-96%, Spe: 35%-55%
[96]	Hospital Israelita Albert Einstein, Brazil	5644 (558)	106 (9)	LR, ANN, SVM, RF, GB	Sen: 95% Spe: 95%, AUC: 95%
[97]	Hospital Israelita Albert Einstein, Brazil	253 (102)	108 (15)	ANN, RF, GBT, Albert LR, SVM	Sen: 68%, Spe: 85%
[98]	Hospital Israelita Albert Einstein, Brazil	599 (81)	108 (16)	SVM	Sen: 70.25%, Spe: 85.98%
[99]	A case series from Wenzhou, Zhejiang, China	53 (-)	23 (-)	LR, KNN, RF, DT, SVM	AUC: 80%
[100]	West London Hospital	398 (-)	Many (-)	CRM, ANN	AUC: 88.1%
[101]	Hospital Sirio Libanes	1945 (545)	40 (40)	DT, RF, GBM, XGBoost, SVM, LGBM, KNN, ANN	Acc: 90%
[102]	Tongji Hospital Affiliated to Huazhong University of Science and Technology	137 (-)	100 (28)	SVM, LR	Acc: 99%
[103]	3000 examples collected at a hospital in Poland	3114 (1941)	- (-)	LR, XGBoost	Sen: 97%, Spe: 11%
[104]	-	3819 (-)	Many (-)	LR, SKLearn, RF, LGBM	-
[105]	Hospital Israelita Albert Einstein, Brazil	5644 (558)	18 (18)	XGBoost, LR	
[106]	UCLA Health System in Los Angeles, California	1455 (182)	Many (27)	RF, LR, SVM, ANN, SGD, XGBoost, ADABoost	Sen: 93%, Spe: 64%
[107]	Three hospitals in the United States, Iran, and Italy	295 (117)	Many (1691)	RF	Sen: 96.1%, AUC: 88.4%
[108]	NYU Langone Health (NYULH)	206,67 (12,47)	Many (-)	LR, XGBoost, MLP, RNN, GRU, LSTM	Sen: 85%, Spe: 86%, AUC: 92%
[109]	Hospitals in the Mount Sinai Health System in New York City	4098 (4098)	Many	XGBoost	AUC: 89%
[110]	A large acute care healthcare system the Mount Sinai Health System	567 (567)	1360 338	RF	AUC: 85.5%
[111]	HM hospitals’ network in Madrid (Spain)	2307 (1696)	29 (25)	LR, RF	Sen: 81.69%, Spe: 81.46%, AUC: 89%
[112]	Patients from eight centers in China, Italy, and Belgium	725 (-)	29 (13)	LR	Acc: 87.5%, Sen: 96.9%, Spe: 88%, AUC: 93%
[113]	Hospitals of Tongji Medical Colleg, Huazhong University of Science and Technology	2520 (-)	53 (34)	LR, SVM, GBDT, ANN	AUC: 97.6%
[114]	Korea Centers for Disease Control and Prevention	3524 (3524)	44 (44)	LR, SVM, KNN, RF, GBDT	AUC: 83%
[115]	Electronic Health Records	86,355 (4759)	Many (-)	LR, XGBoost	AUC: 83.8%
[116]	The ED of an urban multicenter health system	300 (300)	50 (50)	RF	-
[117]	UK Biobank (UKBB data)	465,728 (7846)	97 (15)	XGboost	AUC: 81%
[118]	Hospital Israelita Albert Einstein, Brazil	5644 (279)	106 (97)	RF, LR, XGBoost	Sen: 80%, Spe: 98%
[119]	Hospital Israelita Albert Einstein, Brazil	510 (73)	108 (15)	NB	-
[120]	-	689 (362)	43 (-)	KNN, RF, SVM	ACC: 97.7%
[121]	Painel COVID-19 Estado Do ESPÍRITO Santo	8443 (4826)	Many (-)	LR, LDA, NB, KNN, DT, XGboost, SVM	Sen: 88%, Spe: 82%, AUC: 92%
[122]	Tongji Hospital Affiliated to Huazhong University of Science and Technology	362 (362)	- (-)	RF	Acc: 95%
[123]	Hospital Israelita Albert Einstein, Brazil	5644 (674 )	Many (-)	-	AUC: 94%
[124]	Mass General Brigham (MGB) Healthcare	10,826 (3713)	Many (-)	ANN, SVM	-
[125]	San Raffaele Hospital (Milan, Italy)	207 (105)	Many (-)	LR	-
[126]	24 hospitals in Hong Kong	5148 (-)	Many (-)	-	Sen: 77.8%, Spe:98.3%, AUC: 96.8%
[127]	Tumor Center of Union Hospital affiliated with Tongji Medical College, China	99 (11)	33 (33)	-	AUC: 95.3%
[128]	Tongji Hospital of Wuhan, China	110 (59)	47 (7)	LR	Sen: 98%, Spe: 91%
[129]	World Health Organization	601 (-)	Many (-)	XGBoost	Acc: 100%, Sen: 84.6%, Spe: 84.6%, AUC: 95.3%
[130]	Hospitals in Wuhan, China	294 (208)	15 (-)	RF, SVM	Acc: 84%, Sen: 88%, Spe: 80%
[131]	First Medical Center, Beijing, China	132 (26)	46 (18)	LR, DT, Adaboost	Sen: 100%, Spe: 77.80%
[132]	Hospital Israelita Albert Einstein, Brazil	5644 (520)	17 (17)	DES	Acc: 99.81%, AUC: 99.81%
[133]	National Center for Biotechnology Information Gene Expression Omnibus and European Bioinformatics Institute ArrayExpress	705 (100)	- (-)	LR	AUC: 90%
[134]	Ethics Committee of the Affiliated Yueqing Hospital of Wenzhou Medical University	51 (51)	22 (22)	KELM	Acc: 100%, Sen: 100%, Spe: 100%
[135]	Seven different hospital clinical biomarker datasets from Italy, Brazil, and Ethiopia.	1624 (844)	21 (21)	LDA, XGBoost, RF, LR, KNN	Acc: 91.45%, Sen: 91.44% , Spe: 91.44%
[136]	Keio University Hospital	312 (300)	35 (35)	LR	-
[137]	Wuhan, China	485 (-)	- (-)	XGBoost	Acc: 97%
[138]	San Raffaele Hospital Milan Italia and Hospital Israelita Albert Einstein	5644 (560)	111 (111)	SVM	Acc: 99.29%, Sen: 92.79%, Spe: 100%
[139]	Tongji Hospital of Wuhan, China	375 (201)	300 (3)	XGBoost	Sen: 83%
[140]	Veterans Health Administration Sites, USA	5002 (1079)	68 (54)	RF	Acc: 83.30%, Sen: 83.40% Spe: 89.80%
[141]	Oxford University Hospitals, UK	40,732 (437)	74 (-)	RF, LR, XGBoost	Acc: 92.30%, Sen: 77.40%, Spe: 95.70%

## 4. Features

The importance of features in routine blood tests for COVID-19 diagnosis is significant, because in machine learning and deep learning, these features can be used to build predictive models that can identify patterns in data, allowing for earlier detection and diagnosis of COVID-19. Additionally, routine blood tests can be used to monitor disease progression and treatment response in patients with COVID-19. Changes in these features over time can indicate the severity of the disease and the effectiveness of treatment. Moreover, the identification of the most informative features from routine blood tests can help develop models that can predict disease outcomes and mortality rates. This information can be used to allocate resources and prioritize treatment for patients most at risk of severe illness.

The consequences of a wrong selection of features in medical diagnosis using routine blood tests, especially for COVID-19, can have severe consequences for patient health and treatment outcomes. Importantly, machine learning and deep learning algorithms rely heavily on the selection of appropriate features to make accurate predictions. If the wrong features are chosen, the accuracy and reliability of the diagnostic results can be significantly impacted. In the case of COVID-19 diagnosis, a wrong selection of features could result in misdiagnosis or delayed diagnosis, leading to delayed treatment and potentially worse outcomes for patients. For example, if important features, such as inflammatory markers or lymphocyte counts, are excluded from the model, patients with mild or asymptomatic cases of COVID-19 may be missed, leading to the spread of the disease. Moreover, a wrong selection of features can lead to false positives or false negatives, which can have significant implications for patient care. False positives can lead to unnecessary medical interventions or treatments, which can be costly, time-consuming, and may cause patient harm. On the other hand, false negatives can result in delayed treatment, which can lead to the progression of the disease and worse outcomes for the patient. Additionally, a wrong selection of features can result in a lack of generalizability and poor performance of the model. The model may not perform well on new data or may be specific to a particular population or dataset, limiting its usefulness and applicability in real-world settings.

The blood features that were used by previous studies (see Table 4) are as follows [152]:Hematocrit: The computation of the ratio of erythrocytes (commonly referred to as red blood cells) in the blood is carried out. When this percentage is low, it could signify respiratory difficulties and possibly reveal the severity of COVID-19 cases [153].Hemoglobin: A particular material found in red blood cells carries oxygen in the bloodstream. When someone is diagnosed with pneumonia due to COVID-19, a drop in the level of this material (known as Hb) shortly after the diagnosis could suggest that the pneumonia is getting worse. It is worth noting that anemia is a frequent occurrence in COVID-19 cases as well [154].Red blood cell distribution width (RDW): Another term used to describe this is the RDW coefficient of variation. It offers a way to quantify the variability in the dimensions of erythrocytes; while initially utilized as a diagnostic tool for anemia, it has since evolved into an indicator of infections and more severe ailments, including cardiovascular and cancer. Although it is not a reliable indicator of the presence of COVID-19, it has been recognized as a sign of the disease’s severity, as elevated RDW levels have been associated with mortality in cases of COVID-19 [155].Mean corpuscular hemoglobin (MCH): It pertains to the mean amount of hemoglobin, a protein that carries oxygen throughout the body, present in every individual red blood cell [75]. Sarkar et al. [156] suggested that changes in MCH levels could indicate the presence of COVID-19. A decrease in the MCH value typically signifies a lack of iron in the body, known as iron deficiency anemia. In general, people with COVID-19 tend to display MCH values that are slightly below the normal range, falling within one standard deviation.Mean corpuscular hemoglobin concentration (MCHC): It is a measurement that determines the average hemoglobin concentration inside an individual red blood cell, similar to MCH [75]. A low MCHC value suggests that a person’s red blood cells have insufficient hemoglobin, indicating anemia. According to [157], this metric aided in the differentiation of cases of COVID-19 from pneumonia contracted within the general community. MCV, which stands for Mean Corpuscular Volume, is an indicator of the typical size or volume of erythrocytes [75]. Changes in the mean size of red blood cells, whether an increase or a decrease, can signal underlying health concerns, and research has associated such alterations with the severity of COVID-19 [158].Lymphocytes absolute: A reduced level can be an indication of serious COVID-19, which can prompt early treatment or suggest a negative outcome [154]. This metric serves as a marker for infectious processes.Leukocytes: The immune system’s defensive cells are called white blood cells. Research has demonstrated that COVID-19 has the ability to attack these cells, causing them to discharge pro-inflammatory cytokines that result in an increase in inflammation within the affected individual [159]. Furthermore, it is possible to utilize indicators present in the genetic composition of leukocytes to detect the existence of COVID-19 [160].Basophils absolute: They are crucial cells of the immune system, and their levels tend to rise during prolonged inflammation or allergic reactions. However, research has shown that individuals infected with COVID-19, particularly those with severe cases, experience a significant decrease in basophil counts [161]. Similarly, eosinophils, which play a role in defending the body against parasites and infections, also exhibit reduced levels in COVID-19 patients [162].Platelets: The bone marrow produces these cells that aid in the process of blood coagulation. Keeping a close eye on this measurement is crucial because a rise in its levels may not always be indicative of COVID-19, but can instead point to complications related to the disease, including thrombosis [163].Monocytes absolute: The protection against different microorganisms and viruses is provided by monocytes and macrophages, which are essential constituents of the immune system [164]. Macrophages exist in bodily tissues, whereas monocytes can be found in the bloodstream and are identifiable through blood counts. Despite their beneficial characteristics, these cells can have harmful effects on those with COVID-19, leading to lung infections and lesions. Several studies have revealed a reduction in the number of monocytes in people with COVID-19 [164]. According to other research, such as Meidaninikjeh et al. [165], it is proposed to create novel methods to detect the migration of these cells towards the lungs as a potential sign of COVID-19, and to utilize suitable treatments to reduce lung harm.SARS_CoV2_PCR: It is dependable in verifying the existence of a COVID-19 infection because it detects the virus’s genetic material. The variable under consideration will be allocated a value of 0, which indicates negative instances, and 1, which signifies positive instances.

## 5. Discussion and Analysis

It is challenging to diagnose coronavirus using routine blood testing. Researchers have employed numerous preprocessing strategies, feature extraction approaches, and classification models [166]. Identifying a single strategy or set of methodologies that produce the best outcomes for detecting COVID-19 from regular blood tests is challenging. Most research articles showed accuracy rates of more than 90%, which may be extremely high. Nevertheless, the goal would be to raise the accuracy to about 100%, as inaccurate disease classification, even in a small number of cases, is wholly unacceptable. On the other hand, generalization capacity poses a serious issue for all learning-based methodologies. It results from both the procedures themselves and the diversity of the training dataset. As a result, deep learning has had a significant impact on how routine blood tests are applied, and we anticipate that it will become a more effective methodology in the future [167].

Different techniques of ML and DL have been used in 92 reviewed studies. The ML or DL methods utilized are displayed in Figure 4. Figure 5 shows that with a percentage of 16%, Random Forest is the most used machine learning method, followed by LR (14%), SVM and XGBOOST (11%), KNN and ANN (7%), DT (6%), etc.

Four metrics, including accuracy, sensitivity (recall), specificity, and AUC, were used to diagnose COVID-19 and to evaluate and compare the performance of the suggested methods quantitatively. These four performance metrics used in the literature of COVID-19 diagnosis are shown in Table 5. Accuracy indicates that how many samples are classified properly (ratio of true predictions over all predictions ). Sensitivity or recall refers to the rate of the number of correctly classified COVID-19-positive samples to the total number of suspected samples. Specificity refers to the rate of identifying negative samples correctly. The area under the ROC curve is represented by AUC, from (0, 0) to (1, 1).

A lot of AI knowledge is needed to implement a standard ML-based diagnosis. The most crucial step is identifying the distinctive features that can help with COVID-19 diagnosis. Because of the absence of adequate hardware resources and data availability, the development of DL systems for the diagnosis of COVID-19 has been a difficult endeavor. Although websites such as Google Colab already provide researchers with powerful computing processors, applying and employing these approaches in the real world presents many challenges. While DL and ML models hold great potential for improving COVID-19 diagnosis using routine blood tests, there are still several limitations that must be addressed before they can be used widely in clinical settings: (1) DL and ML models rely on large amounts of diverse data to train and learn from. However, data on COVID-19 are still relatively limited, especially in terms of routine blood tests. This can make it difficult to develop accurate models that apply to different populations and settings. (2) Routine blood tests can be affected by many factors, such as age, sex, underlying medical conditions, and medications. ML models may struggle to account for these factors and may generate inaccurate predictions as a result. (3) ML models are only as good as the data they are trained on and must be carefully validated and tested before they are used in clinical settings. This requires extensive testing and validation of the model on large and diverse datasets. (4) DL models are often described as “black boxes”, because it is difficult to interpret how they arrive at their predictions. This can make it difficult for healthcare providers to understand and trust the predictions generated by these models. (5) ML models used in healthcare are subject to regulatory approval, and there are strict requirements for validation and testing. This can slow down the adoption of these models in clinical settings.

Although we conducted a search in four of the most major databases and found over 4785 documents, there are some drawbacks to this research. The first one is the possibility of selection bias because only English-language research articles were chosen. The keywords used for the queries also influence the results that are obtained. Another limitation of this research is the time frame in which the search was conducted. As with any rapidly evolving field, new research is constantly being published, and the results of this study may not reflect the most recent findings. Additionally, the scope of the search may have been too narrow, focusing only on research articles related to a specific aspect of the topic. Moreover, the quality of the research articles selected in the review may vary, as not all research articles undergo the same level of peer review or scrutiny. It is possible that some of the research articles chosen may have had limitations in their methodology or analysis, which could impact the validity of the conclusions drawn from the review.

We want to take note that the primary benefit of the current study is that it gives the reader a comprehensive list of recent research articles that use various forecasting approaches based on routine blood tests. The reader gets access to a method-based categorization of publications (ML and DL). For anyone interested in this subject, the research articles given in this study would be a good place to start and would hasten their learning in this area of study. For everyone involved in a literature review, the flowchart shown in Figure 3 would be useful. There are two primary drawbacks to this study. We have only considered studies published within the last one to two years. This survey has not covered the material from other captivating databases. This is because of the overwhelming volume of publications that authors would have had to manage.

## 6. Conclusions

Millions of people’s lives have been gravely threatened by the ongoing COVID-19 pandemic in a short amount of time. As the CT scan technique is more expensive and time-consuming than routine blood tests, it is apparent that routine blood tests are more broadly accessible than the CT image dataset. As a result, the majority of researchers used standard blood testing to identify COVID-19. After reviewing the literature in this field, we discover that there is a dearth of annotated data on those impacted by COVID-19. The performance of the aforementioned data-hungry models can be significantly improved by enhancing high-quality datasets of COVID-19 patients. ML and DL can detect the coronavirus using AI techniques when applied to routine blood testing. This study compares some recent research utilizing ML and DL algorithms to detect coronaviruses from routine blood tests obtained from multiple open-source datasets. This study only covers the 92 studies examined, while numerous studies have recently been undertaken based on these findings.

After reviewing 4785 research articles, only 92 were deemed pertinent to the topic under investigation for this study. This can provide the reader with a sense of how uncommon this topic is in the studied field. The application of ML and DL to the prediction of COVID-19 using routine blood tests remains unexplored. It should be noted that 559 full-text research articles were examined for this revision work. In order to appeal to readers, the authors suggest that any upcoming research should take into consideration other databases in the literature review, such as Emerald, Scopus, and Web of Science. In addition, the authors suggest researchers consider using open-source datasets to train and test ML and DL models for predicting COVID-19 using routine blood tests. Open-source datasets can provide a standardized and accessible platform for researchers to develop and test their models. This can also facilitate collaboration and sharing of data and results among researchers in the field. The authors also recommend that future research should explore the potential of using ML and DL in combination with other medical technologies, such as imaging and genomics, to develop more accurate and comprehensive diagnostic tools for COVID-19. By leveraging multiple sources of data, researchers can develop more holistic approaches to predicting and treating the disease.

## Figures and Tables

**Figure 1 diagnostics-13-01749-f001:**
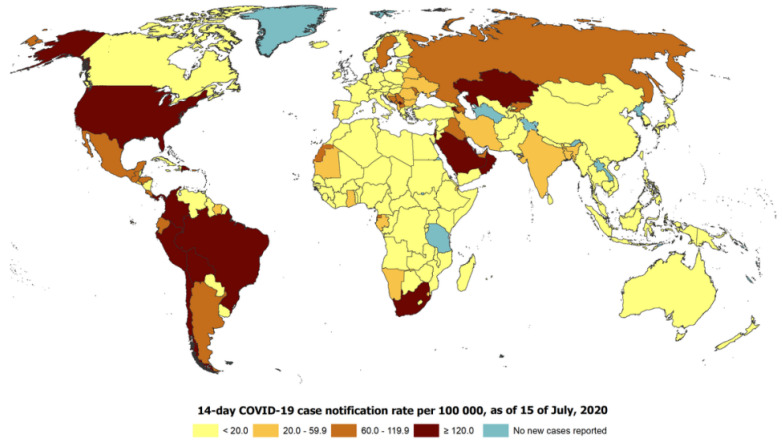
The global 14-day COVID-19 case notification rate per 100,000 as of 15 July 2020.

**Figure 2 diagnostics-13-01749-f002:**
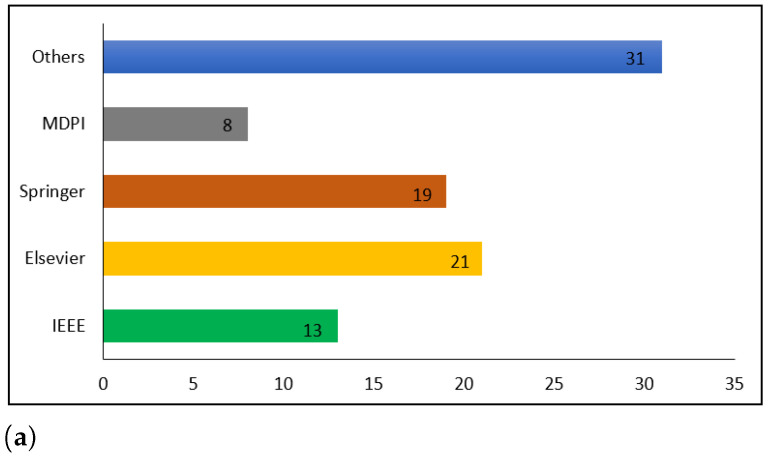
Statistics on ML and DL publications in COVID-19. (**a**) Publication per database, (**b**) COVID-19 publication, (**c**) Distribution of the published COVID-19 research articles, (**d**) COVID-19 publication.

**Figure 3 diagnostics-13-01749-f003:**
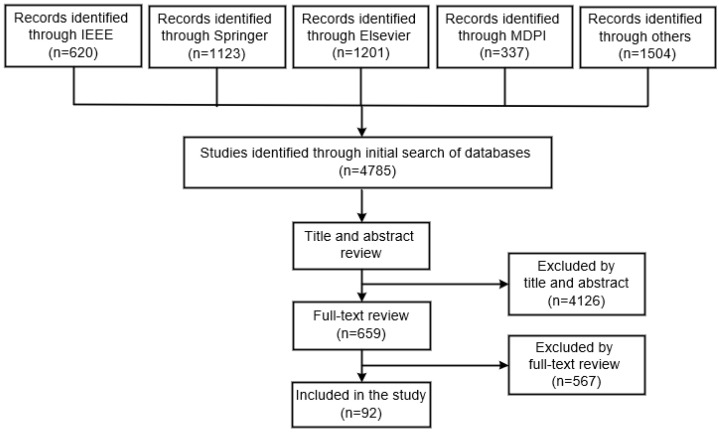
Summary of the research article selection process.

**Figure 4 diagnostics-13-01749-f004:**
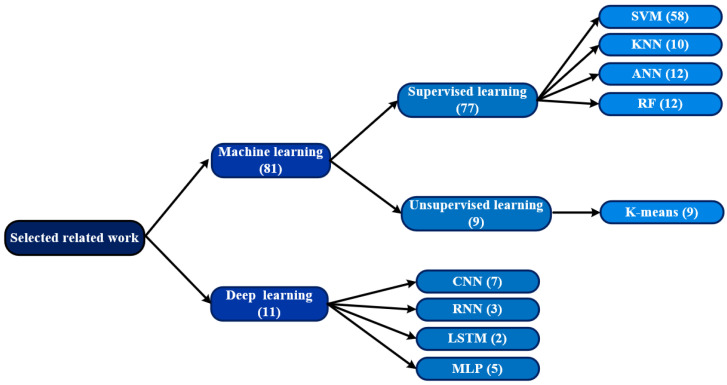
The most frequently used techniques for COVID-19 detection among the studies included in this survey. The numbers in parentheses indicate the number of research articles.

**Figure 5 diagnostics-13-01749-f005:**
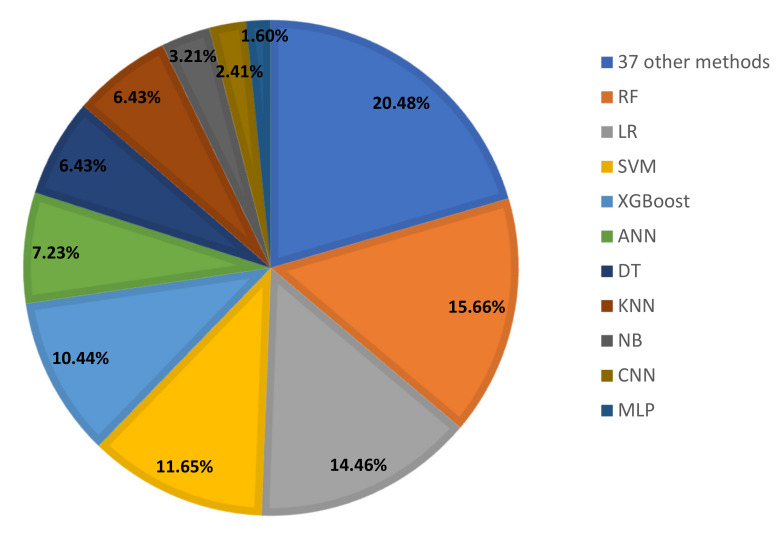
ML and DL techniques used for COVID-19 diagnosis.

**Table 1 diagnostics-13-01749-t001:** Research articles identified in the databases under investigation.

Database	Number of Research Articles
IEEE	620
Springer	1123
Elsevier	1201
MDPI	337
Others	1504
Total	4785

**Table 3 diagnostics-13-01749-t003:** List of DL models that diagnose COVID-19.

Ref.	Dataset Source	Dataset Size (COVID-19)	Total Features (Selected)	Model Used	Metric Results
[142]	Hospital Israelita Albert Einstein, Brazil	5644 (559)	18 (18)	VAE-based SVM	AUC: 99.60 %
[66]	Hospital Israelita Albert Einstein, Brazil	5644 (652)	50 (17)	ANN, KNN	Acc: 89.25%
[143]	-	600 (520)	20	ANN, CNN, RNN	Acc: 94.95%, AUC: 100%
[144]	Hospital Israelita Albert Einstein, Brazil	5644 (559)	18 (18)	ANN, CNN, LSTM, RNN	ACC: 86.66%, AUC: 62.5%
[145]	Tongji Hospital Affiliated to Huazhong University of Science and Technology	181 (181)	Many (56)	ANN	AUC: 96.8%
[146]	Hospital Israelita Albert Einstein, Brazil	5644 (559)	18 (18)	CNN, RF, ANN	Acc: 92.52%
[147]	Hospital Israelita Albert Einstein, Brazil, San Raffaele Hospital in Italy	7360 (1374)	Many (-)	DNN, KNN	AUC: 88%
[148]	From 18 medical centers in China.	905 (419)	-	CNN, LR, SVM, MLP	AUC: 92%
[149]	Hospital Israelita Albert Einstein, Brazil	5644 (559)	18 (15)	Deep forest	Acc: 99.5%, Sen: 95.28%, Spe: 99.96%
[150]	Hospital Israelita Albert Einstein, Brazil	600 (-)	18 (-)	CNN + GRU, CNN + Bi-RNN, CNN + Bi-LSTM, CNN + Bi-GRU	Acc: 94.15%, AUC: 91.00%
[151]	Hospital Israelita Albert Einstein, Brazil	-(-)	-(-)	CNN, KNN, NB, DT	Acc: 80%

**Table 4 diagnostics-13-01749-t004:** List of blood features selected by previous research articles.

Feature	Refs.	Number of Research Articles
Hematocrit	[9,16,64,65,66,67,72,77,79,81,82,83,87,88,89,96,98,111,112,126,129,131,134,138,139,144,151]	28
Hemoglobin	[9,16,64,66,67,79,83,86,87,98,111,122,134,139,151]	15
RDW	All	92
MCH	[9,64,67,68,77,79,111,131,138,144]	10
MCHC	[9,16,20,65,66,67,68,72,73,74,77,78,79,82,83,86,88,89,90,94,96,98,101,102,106,112,118,120,122,126,128,129,131,132,134,135,137,138,139,144,146]	41
Lymphocytes absolute	All	92
Leukocytes	[9,67,111,126,141]	5
Basophils absolute	[65,66,67,68,72,78,79,86,87,89,95,111,112,129,131,137,144,150]	18
Platelets	All	92
Monocytes absolute	[67,72,77,79,82,87,88,90,96,98,106,111,112,126,129,131,134,137,138,139]	20
SARS_CoV2_PCR	[66,67,68,88,98,111,128,131,134,135,141]	11

**Table 5 diagnostics-13-01749-t005:** Performance metrics used in the research articles.

Performance Metrics	Refs.	Number of Research Articles
Accuracy	[9,16,20,64,65,66,70,72,75,76,78,81,82,83,84,87,89,92,101,102,112,120,122,129,130,132,134,135,137,138,140,141,143,144,146,149,150,151]	39
Sensitivity	[9,20,64,67,68,72,73,74,75,76,77,78,79,83,84,86,88,90,92,94,95,96,97,98,103,106,107,108,111,112,118,121,126,128,129,130,131,134,135,138,139,140,141,149]	44
Specificity	[9,64,72,73,74,75,76,77,78,79,83,84,86,88,90,92,94,95,96,97,98,103,106,108,111,112,118,121,126,128,129,130,131,134,138,140,141,149]	38
AUC	[9,67,68,69,70,72,73,74,75,77,78,80,81,83,85,86,87,90,91,92,93,94,96,99,100,107,108,109,110,111,112,113,114,115,117,121,123,126,127,129,132,133,142,143,144,145,147,148,150]	49

## Data Availability

Data will be made available on request.

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
