# Peer review of "A Survey of COVID-19 Diagnosis Using Routine Blood Tests with the Aid of Artificial Intelligence Techniques"

_diagnostics, 2023, doi:10.3390/diagnostics13101749_

Round 1

Reviewer 1 Report

1. The authors should use the same format for the terms in Line 79, like COVID-19. They should look into the entire manuscript and double check.

2. The figure 2 is not matching the descriptions in the main text, which should the flowchart. The figure itself is Statistics on ML and DL publications.

3. Overall, this survey will help the readers learn state of art research progress in this field, and the authors screen all the relevant publications to support their survey. However, the analysis and discussion are not sufficient in the Result and discussion section. They should give more detailed analysis and comparison on DL and ML, how those models improve the diagnosis of COVID-19, how much they could improve, what their advantages and disadvantages, and what are the limitations of DL and ML, etc... Not just list everything in the tables. 

Author Response

Dear editor,

We thank the reviewer for their thorough reviews which helped with the paper improvement. We have revised our paper to address the reviews’ comments. Detailed answers are available below for each comment.

  1.  The authors should use the same format for the terms in Line 79, like COVID-19. They should look into the entire manuscript and double check.
    Ans: We checked and corrected all of them.
  2. The figure 2 is not matching the descriptions in the main text, which should the flowchart. The figure itself is Statistics on ML and DL publications.
    Ans: We rewrited Section 2 and added the selection process of articles in Figure 2.
  3.  Overall, this survey will help the readers learn state of art research progress in this field, and the authors screen all the relevant publications to support their survey. However, the analysis and discussion are not sufficient in the Result and discussion section. They should give more detailed analysis and comparison on DL and ML, how those models improve the diagnosis of COVID-19, how much they could improve, what their advantages and disadvantages, and what are the limitations of DL and ML, etc... Not just list everything in the tables.
    Ans: We extended Section 5 (Discussion and analysis). We added a discussion about ML and DL models, also the limitations of this survey. Moreover, we presented the future work in the conclusion section.

Reviewer 2 Report

I am really grateful to review this manuscript. In my opinion, this manuscript can be published once some revision is done successfully. This study used 112 references to review artificial intelligence-based diagnosis of COVID-19 with routine blood tests. I would argue that this is a great achievement. However, explainable artificial intelligence is gaining great attention now hence it would be really helpful to add information on important predictors either as an additional column in Table 2 or a separate table like Table 4. 

Minor editing of English language required 

Author Response

Dear editor,

We thank the reviewer for their thorough reviews which helped with the paper improvement. We have revised our paper to address the reviews’ comments. Detailed answers are available below for each comment. 

  1. I am really grateful to review this manuscript. In my opinion, this manuscript can be published once some revision is done successfully. This study used 112 references to review artificial intelligence-based diagnosis of COVID-19 with routine blood tests. I would argue that this is a great achievement. However, explainable artificial intelligence is gaining great attention now hence it would be really helpful to add information on important predictors either as an additional column in Table 2 or a separate table like Table 4.
    Ans: One of the most predictors is blood features. We added section 4 and explain about the important features selected in previous articles. 

Reviewer 3 Report

The article entitled “A survey of COVID-19 diagnosis using routine blood tests with the aid of artificial intelligence techniques” is well-written and, from my point of view, would be of interest for the readers of Diagnositcs. In spite of this and before its publication, I would like to suggest the following changes:

In section “2. Protocol for choosing COVID-19-related articles” I would recommend researchers to cite PRISMA for systematic reviews and also, clarify how was the keywords search performed. Please note that in many articles a flowchart indicating the number of articles considered in each stage is employed. I suggest authors to consult the flowchart in the work:

Todorov, I.B.; Sánchez Lasheras, F. Forecasting Applied to the Electricity, Energy, Gas and Oil Industries: A Systematic Review. Mathematics 2022, 10, 3930. https://doi.org/10.3390/math10213930

Figure 2: each sub-figure should have their own caption and please also note that numbers in b) and d) are very difficult to read.

Figure 3 that categorizes related Works would be improved in the number of works in each node is included.

I suggest deleting Table 1 and introducing the abbreviations in the text the first time that they are employed.

Figure 4: please think about changing colours in order to make figure more easy to interprete.

No problem at all with English.

Author Response

Dear editor,

We thank the reviewer for their thorough reviews which helped with the paper improvement. We have revised our paper to address the reviews’ comments. Detailed answers are available below for each comment. 

  1. In section “2. Protocol for choosing COVID-19-related articles” I would recommend researchers to cite PRISMA for systematic reviews and also, clarify how was the keywords search performed. Please note that in many articles a flowchart indicating the number of articles considered in each stage is employed. I suggest authors to consult the flowchart in the work:
    Todorov, I.B.; Sanchez Lasheras, F. Forecasting Applied to the Electricity, Energy, Gas and Oil Industries: A ´ Systematic Review. Mathematics 2022, 10, 3930. https://doi.org/10.3390/math10213930
    Ans: We got an idea from the given reference and modified section 2 accordingly.
  2. Figure 2: each sub-figure should have their own caption and please also note that numbers in b) and d) are very difficult to read.
    Ans: We broke Figure 2 in to four sub-figures and made a caption for every sub-figure. We changed the color of numbers.
  3. Figure 3 that categorizes related Works would be improved in the number of works in each node is included.
    Ans: We changed it.
  4.  I suggest deleting Table 1 and introducing the abbreviations in the text the first time that they are employed.
    Ans: The most of the abbreviations are from table 2 and 3. By adding the full name of the abbreviations, the tables cross the margin. We added the abbreviation list at the end of the article based on the MDPI format.
  5. Figure 4: please think about changing colours in order to make figure more easy to interprete.
    Ans: We changed it. 

Round 2

Reviewer 1 Report

all the concerns have been addressed, and it can be accepted for publication with current version